# Investigation of the Effects of 3D Printing Parameters on the Mechanical Properties of Bone Scaffolds: Experimental Study Integrated with Artificial Neural Networks

**DOI:** 10.3390/bioengineering12030315

**Published:** 2025-03-19

**Authors:** Rixiang Quan, Sergio Cantero Chinchilla, Fengyuan Liu

**Affiliations:** School of Electrical, Electronic and Mechanical Engineering, University of Bristol, Bristol BS8 1TR, UK; rixiang.quan@bristol.ac.uk

**Keywords:** 3D printing, artificial neural network (ANN), bioengineering, bone scaffolds, machine learning, mechanical properties, printing parameters

## Abstract

Scaffolds are critical in regenerative medicine, particularly in bone tissue engineering, where they mimic the extracellular matrix to support tissue regeneration. Scaffold efficacy depends on precise control of 3D printing parameters, which determine geometric and mechanical properties, including Young’s modulus. This study examines the impact of nozzle temperature, printing speed, and feed rate on the Young’s modulus of polylactic acid (PLA) scaffolds. Using a Prusa MINI+ 3D printer (Prusa Research a.s., Prague, Czech Republic), systematic experiments are conducted to explore these correlations. Results show that higher nozzle temperatures decrease Young’s modulus due to reduced viscosity and weaker interlayer bonding, likely caused by thermal degradation and reduced crystallinity. Printing speed exhibits an optimal range, with Young’s modulus peaking at moderate speeds (around 2100 mm/min), suggesting a balance that enhances crystallinity and bonding. Material feed rate positively correlates with Young’s modulus, with increased material deposition improving scaffold density and strength. The integration of an Artificial Neural Network (ANN) model further optimized the printing parameters, successfully predicting the maximum Young’s modulus while maintaining geometric constraints. Notably, the Young’s modulus achieved falls within the typical range for cancellous bone, indicating the model’s potential to meet specific clinical requirements. These findings offer valuable insights for designing patient-specific bone scaffolds, potentially improving clinical outcomes in bone repair.

## 1. Introduction

Bone scaffolds are critical components in regenerative medicine, particularly in bone tissue engineering, where they act as temporary structures mimicking the extracellular matrix of bone to facilitate the regeneration of damaged or diseased tissue [1]. The effectiveness of these scaffolds is highly dependent on the precise control of 3D printing parameters, which significantly impact both their geometrical properties—such as filament thickness and dimensional accuracy—and their mechanical properties, including Young’s modulus [2]. Understanding the relationship between these printing parameters and scaffold performance is essential for optimizing scaffold design and ensuring their efficacy in clinical applications [3,4]. Scaffolds guide new tissue growth while maintaining the necessary space for bone repair, playing a crucial role in the healing process. Designed to gradually degrade within the body, they synchronize with natural bone regeneration, eliminating the need for surgical removal. The clinical advantages of bone scaffolds are considerable—they can expedite the healing process, reduce the necessity for multiple surgeries, and improve overall outcomes in bone repair procedures [5]. Additionally, the ability to customize these scaffolds to meet patient-specific anatomical requirements through advanced manufacturing techniques further enhances their therapeutic potential [6,7].

Polylactic acid (PLA) has emerged as a preferred material for fabricating bone scaffolds due to its favorable properties. PLA is both biocompatible and biodegradable, making it suitable for medical applications where the material can safely degrade via hydrolysis in the body over time [5,6]. With a relatively low melting point of 144 °C [8], PLA is easy to process, particularly in 3D printing, making it an attractive option for creating scaffolds that support bone growth and gradually degrade as new tissue forms. PLA was chosen over other polymers like polycaprolactone (PCL) due to its superior mechanical properties, including higher tensile strength and modulus, which are critical for the mechanical stability required in scaffold applications [8]. The adoption of 3D printing technologies in scaffold fabrication offers significant benefits, such as the ability to produce complex geometries and patient-specific designs [9]. For instance, scaffolds with circular pore structures have been found to exhibit superior fatigue resistance due to more uniform mechanical stress distribution and reduced stress concentration, making this geometry particularly advantageous for long-term bone repair applications [10]. Three-dimensional printing allows for precise control over scaffold architecture, ensuring optimal porosity, mechanical strength, and biological performance, which is especially valuable in producing scaffolds tailored to individual patient needs, thereby enhancing the effectiveness of bone tissue engineering [6,11].

Despite these advantages, using PLA in 3D printing bone scaffolds presents challenges. A primary concern is how printing parameters—such as nozzle temperature, printing speed, and material feed rate—affect the mechanical properties and dimensional accuracy of scaffolds [12]. These parameters significantly influence interlayer bonding, porosity, and overall mechanical performance, which can impact the scaffold’s effectiveness in supporting bone regeneration [2]. The material properties of PLA, including its melting temperature, crystallinity, and thermal stability, are crucial to its performance as a scaffold material. The degree of crystallinity in PLA, which affects its mechanical properties and structural integrity, is particularly important. The crystallization process during 3D printing, especially the formation of different crystalline forms (such as the α and δ forms), is directly influenced by printing parameters, including build-platform temperature. This crystallization behavior affects the viscosity of PLA during extrusion, the cooling and solidification rates, and the mechanical properties of the final scaffold [13]. Therefore, precise control of the printing process is essential to optimize crystallinity and achieve the desired scaffold characteristics, including mechanical strength and degradation behavior.

To refine 3D printing parameter optimization, machine learning, particularly ANNs, has proven effective in predicting and optimizing scaffold properties [14]. Machine learning models can analyze large datasets of experimental results and identify complex relationships between printing parameters and scaffold performance that might not be immediately apparent through traditional methods [15]. In this study, an ANN model was integrated to predict scaffold geometrical properties (height, width) and mechanical performance (Young’s modulus), enabling more precise control over the fabrication process and ensuring that key mechanical and dimensional targets are met. The application of machine learning allows for the rapid evaluation of different printing parameter combinations, improving scaffold design efficiency and aiding in the development of patient-specific solutions for bone tissue engineering.

This research focuses on exploring the effects of nozzle temperature, printing speed, and material feed rate on the Young’s modulus and dimensional accuracy of PLA scaffolds. The current understanding of how these parameters interact with the material properties of PLA is limited, necessitating systematic research to lay the groundwork for future optimization efforts. By determining the optimal printing conditions, the performance of PLA scaffolds in bone tissue engineering applications could be significantly improved. The key contributions of this study include (1) the exploration of relationships between printing parameters and both geometric and mechanical properties of PLA scaffolds, (2) the proposal of simple models that relate these key parameters to mechanical and geometric properties, and (3) the integration of an ANN, to predict and optimize scaffold properties based on printing parameters. Ultimately, the findings of this research could contribute to the development of more effective and reliable regenerative treatments, enhancing patient outcomes in bone repair and regeneration.

This study is structured as follows: Section 2 covers the experimental setup and ANN model development, Section 3 analyzes parameter effects and ANN optimization, and Section 4 summarizes findings and future improvements.

## 2. Materials and Methods

The effect of the printing parameters on the geometric and mechanical properties of PLA scaffolds has been addressed through experimental testing and statistical analysis. An overview of this process is shown in Figure 1. Initially, printing parameters and materials for the experiment were selected, following the printing process. To evaluate the geometric and mechanical properties of PLA bone scaffolds, Electron Microscopy (SEM) and compression mechanical tests were used to assess both the filament and scaffold geometry, as well as the strength of the specimens.

The final step involves analyzing the correlation between the printing parameters, filament thickness, width, scaffold dimensions, and their mechanical properties, particularly the resulting Young’s modulus.

### 2.1. Materials

Polylactic acid (PLA) “Ecogenius” 3D printing filament purchased from Sigma-Aldrich (Merck Group, Rahway, NJ, USA) was used to produce scaffolds. PLA is a biodegradable semi-crystalline thermoplastic with a density of 1.24 g/cm^3^, a melt flow index (MFI) of 6.0 g/10 min, a filament diameter of 1.75 mm, a melting point of approximately 144 °C, Vicat softening temperature of approximately 60 °C, tensile strength of 65.5 MPa and tensile modulus of 3.31 GPa machine Direction (MD), and 3.86 GPa transverse Direction (TD) [16].

### 2.2. Scaffold Design and Fabrication

The scaffold model used in the experiments features a lattice structure composed of parallel filaments arranged in a layer-by-layer fashion, as shown in Figure 2. The initial layer consists of 10 vertically aligned parallel filaments. In subsequent layers, the filaments are aligned horizontally, perpendicular to the previous layer, creating a grid-like architecture. This alternating pattern is maintained across 20 layers, which is crucial for facilitating cellular integration and tissue growth—key factors for successful bone tissue engineering. Studies have shown that such a configuration optimizes interconnectivity and pore size, enhancing osteoconduction and vascularization [17]. The average pore size achieved with this configuration was approximately 400 microns, as measured from the CAD model, which falls within the optimal range for bone ingrowth and vascularization identified in tissue engineering research [18,19]. This pore size effectively supports cell attachment and proliferation, essential for bone tissue regeneration [20]. Notably, scaffolds with 30% porosity (P% in Equation (1)) have demonstrated similar cell proliferation performance to those with 50% porosity, indicating a balance between structural integrity and biological functionality [21]. The scaffold in this study had a porosity of around 51.1%, as measured from the CAD model.

In this study, a modeling-based simulation method was used to calculate the porosity of the scaffolds. The porosity was determined by first measuring the filament dimensions and using these values to construct a CAD model of the scaffold.(1)P%=1−VsVt×100%
where $V_{s}$ represents the volume of solid material in the scaffold (the volume occupied by the printed PLA material), and $V_{t}$ is the total volume of the scaffold, including both the solid material and the void spaces.

Key parameters defining this structure include a filament diameter (FD) of 0.4 mm, layer thickness (LT) of 0.3 mm, filament gap (FG) of 0.6 mm, and a total of 20 layers (LN). These dimensions were consistently maintained across all prints to ensure manufacturing uniformity.

The Prusa MINI+ 3D printer, used for scaffold fabrication, is a commercial-grade device with a compact design [22]. Its size and functionality make it ideal for future applications, allowing for easy adoption and operation in various settings. With a 0.4 mm nozzle, this printer delivers the precision and adaptability needed to produce the complex scaffold geometries crucial for biomedical applications [22,23]. In this study, the nozzle diameter was kept constant to isolate the effects of nozzle temperature, printing speed, and material feed rate on the mechanical properties of the scaffolds, ensuring a focused investigation into these key parameters.

### 2.3. Design of Experiments

The parameters investigated included printing temperature (PT: 180 °C to 250 °C), printing speed (PS: 100 mm/min to 5100 mm/min), and material feed rate (MFR: 15% to 42%), as shown in Table 1. These ranges were chosen to cover the operational capabilities of the printer and the thermal properties of PLA, ensuring that the effects of both lower and upper limits on scaffold integrity and performance were thoroughly tested. The selected temperature range considers PLA’s thermal degradation point and optimal extrusion temperature, while the speed and material feed rate ranges allow for evaluation under varying flow rates and cooling times [24].

In this study, material feed rate (MFR) is defined as the ratio between the nozzle movement speed and the material extrusion speed. When the filament diameter matches the nozzle diameter and the material is extruded at the same rate as the nozzle movement (e.g., a nozzle moving at 1 mm/s with a filament feed rate of 1 mm/s), the material feed rate is considered 100%. However, in cases where over-extrusion or under-extrusion is desired, the material feed rate can be adjusted accordingly to either increase or decrease material deposition relative to nozzle movement.

To efficiently manage the extensive parameter space (15 temperature settings, 11 speed settings, and 10 material feed rate settings, leading to 630 possible combinations), the study employed a Central Composite Design (CCD) approach. Instead of testing all possible combinations, each parameter was examined independently by fixing the other two parameters. For instance, when assessing the impact of printing temperature, the printing speed and material feed rate were fixed at specific values (e.g., PS at 800 mm/min and FR at 30%). This structured approach enabled a focused exploration of each parameter’s impact while maintaining a manageable number of experiments. Three replicate samples were prepared for each parameter combination, for a total of 108 samples, as detailed in Table 1, to evaluate the specific effects of these parameters on the mechanical properties and geometric structure of the scaffolds.

### 2.4. Morphological Characterization

Scanning Electron Microscopy (SEM) was performed using a TM3030Plus tabletop microscope (Hitachi, Tokyo, Japan) to examine the morphology and surface characteristics of the 3D-printed scaffolds. The analysis was conducted at 5 kV without any coating, allowing for a clear visualization of the scaffold structure. For quantitative analysis, a custom Python 3.8 script was used to precisely measure the fiber diameter and pore size across various sections of the scaffold. The script calculates the filament width in vertical direction from P2 to the line formed by points P4 and P5 (Figure 3A), and similar measurements are taken for P6 to line P8–P9 and P10 to line P12–P13, with the average of these distances used to determine the filament width. The same approach is applied to measure the layer height in the side view, where the script calculates the vertical distance from point P1 to the line formed by points P3 and P4 (Figure 3B) and repeated for P5 to line P7–P8 and P9 to line P11–P12; this method enabled robust statistical analysis, with at least 20 measurements taken for each scaffold. For each set of measurements, both mean values and 5–95 percentiles were calculated to assess central tendencies and the range of variability.

### 2.5. Mechanical Characterization

The mechanical properties of the fabricated scaffolds were systematically evaluated to ensure the reliability and reproducibility of the results. Mechanical compression tests were conducted using the INSTRON 3344 testing system (Norwood, MA, USA) equipped with a 2000 N load cell. The scaffolds, which were block-shaped with approximate dimensions of 9.4 mm in length and width and 6.0 mm in height ($h_{0}$), were tested to assess their mechanical properties.

To ensure robust data for statistical analysis, three scaffold samples from each set of printing parameters were tested. These tests were performed on dry scaffolds, applying a constant compression rate of 1 mm/min up to a maximum strain value of 0.85. During these uniaxial compression tests, the system software continuously recorded the force (F) and corresponding displacement values. The engineering stresses (σ) and strains (ε) were computed using Equations (2) and (3) [25]:(2)δ=FA
where A represents the initial sample cross-sectional area.(3)ε=∆hh0
where ∆h denotes the variation in scaffold height and $h_{0}$ is the original height of the scaffold block.

### 2.6. Machine Learning Model Development and Optimization

In this study, an ANN was employed to predict the geometric properties (height, width) and mechanical performance (Young’s modulus) of 3D-printed bone scaffolds based on key 3D printing parameters: printing temperature, printing speed, and material feed rate.

Prior to model training, the input features were standardized, which transformed the data into a distribution with a mean of 0 and a standard deviation of 1. This scaling ensures that all variables, regardless of their original range, are normalized to prevent any specific feature from disproportionately influencing the model’s predictions.

The ANN architecture comprised a single hidden layer with 64 neurons and a ReLU activation function. To prevent overfitting, a Dropout layer with a rate of 0.2 was applied after the hidden layer. The output layer consisted of a single neuron with a linear activation function, which is suitable for regression tasks that predict continuous values. The model was compiled using the Adam optimizer with a learning rate of 0.001 and Mean Squared Error (MSE) was chosen as the loss function to minimize the error between predicted and actual values.

The dataset (Section 2.3) was divided into a training set (80%) and a testing set (20%), with specific indices manually selected for testing. Three separate ANN models were trained, each responsible for predicting scaffold height, width, and Young’s modulus. The models were trained for 200 epochs with a batch size of 8, ensuring that the model had sufficient iterations to learn the underlying patterns in the data.

The ANN model training was conducted on a workstation equipped with an Intel Core i7-12700K CPU and an NVIDIA RTX 3090 GPU, running on Windows 10. The model was implemented using Python 3.11.7 with TensorFlow 2.17.0 and Keras 3.5.0. Additional dependencies included NumPy 1.26.4, Pandas 2.1.4, and Scikit-learn 1.5.1.

Following the model training, a grid search was conducted to identify the optimal combination of printing parameters that would maximize Young’s modulus while ensuring that the predicted scaffold height and width remained within predefined limits (0.35–0.45 mm which has porosity around 42.1% to 59.1%) for both dimensions. This grid search systematically evaluated various combinations of printing temperature, printing speed, and material feed rate, using the trained ANN models to predict the resulting properties for each combination. The combination that produced the highest Young’s modulus while meeting the geometric constraints was selected as the optimal set of parameters.

## 3. Results and Discussion

A one-way analysis of variance (One-way ANOVA) was employed to assess the effects of printing parameters (nozzle temperature, printing speed, and material feed rate) on key outputs including filament thickness, filament width, and compressive Young’s modulus. The results revealed statistically significant effects of all parameters, as shown in Table 2.

A total of 36 experimental parameter sets were applied across 108 samples (with three samples per set) to ensure a comprehensive analysis. In addition to dimensional measurements, compression tests were systematically conducted to determine the Young’s modulus for each scaffold, providing a quantitative measure of the material’s elasticity. The results from these tests were correlated with the scaffolds’ dimensional and structural features to understand how these properties influence mechanical behavior and performance. Additionally, an ANN model was employed to further enhance the analysis, predicting scaffold properties such as height, width, and Young’s modulus based on the input 3D printing parameters. The ANN model provided optimized predictions, enabling the identification of the ideal printing conditions for maximizing mechanical strength while ensuring geometric constraints were met.

### 3.1. Layer Height

Proper adjustment of layer height is crucial for achieving the desired porosity and interlayer adhesion, both of which significantly impact cell infiltration and nutrient diffusion throughout the scaffold. Optimal layer heights enhance the scaffold’s mechanical stability while promoting new tissue formation and integration.

#### 3.1.1. Influence of Nozzle Temperature on Layer Height

Figure 4A demonstrates a linear relationship between nozzle temperature and layer height in the 3D printing of PLA scaffolds. The layer height generally increases with temperature, indicating that higher temperatures may facilitate smoother extrusion and better filament formation, leading to slightly increased layer heights. This trend is likely due to the improved fluidity of PLA at elevated temperatures, which reduces viscosity and produces thicker layers. These findings are consistent with previous studies [2,24], which observed that higher temperatures reduce PLA viscosity, thereby enhancing the extrusion process and increasing layer heights while improving surface smoothness [12]. This relationship is particularly beneficial in the design of large-scale or complex bone repair scaffolds, where higher temperatures can achieve greater layer heights. Additionally, the smoother surface finish attained at higher temperatures is advantageous for implants requiring optimal integration with surrounding tissues [2].

#### 3.1.2. Impact of Printing Speed on Layer Height

In Figure 4B, layer height exhibits a decreasing trend with an increase in printing speed. Starting from a layer height of approximately 0.4893 mm at 100 mm/min, it decreases to about 0.3140 mm at 5100 mm/min. This trend can be explained by the reduced time for the extruded material to settle before solidifying at higher speeds, leading to thinner layers. The decrease in layer height with increased speed is significant, indicating that faster printing speeds compromise layer thickness, possibly affecting the structural integrity of the printed object [26].

#### 3.1.3. Effect of Material Feed Rate on Layer Height

Figure 4C shows that increasing the material feed rate from 15% to 42% results in a clear upward trend in layer height. This suggests that a higher material feed rate, which pushes more material through the nozzle per unit time, directly contributes to increased layer thickness. Consequently, material feed rate can directly influence the porosity of the scaffold, as higher material deposition reduces the space between the printed layers, potentially leading to a denser, less porous structure.

Figure 5 illustrates the influence of 3D printing parameters on the layer height of PLA scaffolds, showing that material feed rate and nozzle temperature play the dominant role in determining the layer height. The results suggest that to achieve the nominal design layer height of 0.4 mm (FD = 0.4 mm), it is crucial to adjust both the nozzle temperature and the material feed rate within optimal ranges.

For instance, when aiming for a consistent layer height close to 0.4 mm, the material feed rate should be set to around 25–30%, with the nozzle temperature maintained between 200 °C and 220 °C, and kept constant around 2500–3500 mm/min; these conditions help achieve a stable layer height while minimizing deviations from the target thickness.

### 3.2. Filament Width

Similar to layer height, the width of the extruded filaments plays a critical role in determining the scaffold’s pore size and overall geometry. Controlling filament width is essential to create precise and consistent pore architectures that support cell adhesion, proliferation, and vascularization [5].

#### 3.2.1. Influence of Nozzle Temperature on Filament Width

In Figure 6A, the analysis of filament width as a function of nozzle temperature shows an almost constant trend with only minor variations. Higher temperatures may improve the flow of PLA material, potentially leading to thicker filament formation. However, the observed trend indicates that this effect is not pronounced. This could be due to a balance between increased material flowability and changes in cooling rates and material viscosity, which counteract significant increases in filament width. According to the literature, while higher nozzle temperatures reduce material viscosity and can lead to better flow and potentially thicker filaments, the relationship is not straightforward. Factors such as the cooling rate and the behavior of the material as it transitions from a molten to a solid state play a crucial role in determining the final filament width [27].

#### 3.2.2. Impact of Printing Speed on Filament Width

Figure 6B shows that filament width seems to increase with the increase in printing speed up to a peak width at 4100 mm/min, after which it slightly decreases. This trend could imply that, at moderate speeds, the material has sufficient time to spread out before solidifying, leading to broader filament [28]. However, at very high speeds, the rapid movement might not allow the material to settle and spread as much, leading to a slight reduction in width. This indicates an optimal speed range where filament width is maximized, potentially beneficial for applications requiring wider filament extrusion.

#### 3.2.3. Effect of Material Feed Rate on Filament Width

The relationship between material feed rate and filament width demonstrates a clear positive correlation in Figure 6C. This direct relationship can be attributed to the increased volume of material being extruded per unit time at higher material feed rates, leading to thicker filament extrusion. It has been found that the maximum extrusion width can lead to enhanced compressive performance. These parameters strengthen the scaffold due to better layer adhesion and stronger bonds at the interconnection of struts, thereby improving load-bearing capacity. This enhancement is crucial for applications where mechanical stability under physiological loads is required [29].

Figure 7 compares the effect of 3D printing parameters on filament width, highlighting that material feed rate plays the dominant role in determining the filament width. The data indicate that to achieve the nominal design width of 0.4 mm, it is crucial to adjust the material feed rate appropriately. Specifically, reducing the material feed rate to around 20% is necessary to achieve this target width, particularly when the nozzle temperature and printing speed are kept within typical ranges (e.g., 200–220 °C for nozzle temperature and 2000–3000 mm/min for printing speed).

In cases where the material feed rate was kept constant at 30%, as shown in previous experiments (Figure 6), the filament width consistently exceeded the design specification, underscoring the importance of material feed rate adjustments in controlling filament dimensions. Therefore, for applications requiring precise filament width, such as in highly detailed or structurally critical scaffolds, reducing the material feed rate is a key adjustment. By carefully selecting a lower material feed rate, along with optimal nozzle temperature and printing speed, it is possible to achieve more accurate filament widths that meet specific design requirements.

### 3.3. Dimensional Precision Ratio (DPR)

In order to assess the overall geometrical precision of the 3D printing process, the Dimensional Precision Ratio (DPR) is utilized. DPR measures the accuracy and reproducibility of the scaffold dimensions compared to the designed model. High DPR values indicate a 3D printing process that can reliably replicate intricate designs, which is crucial for mimicking the complex structure of native bone [30]. This precision is important for the scaffold to fit anatomical sites accurately and function effectively.

The Dimensional Precision Ratio (DPR) can be defined mathematically as:(4)$$DPR=\frac{p_{measured\;}}{p_{CAD}}$$
where $p_{measured}$ is the measured dimension of the scaffold, and $p_{CAD}$ is the designed dimension from the CAD model.

In Figure 8A, the Dimensional Precision Rate (DPR) shows a slight upward linear trend as the nozzle temperature increases from 180 °C to 250 °C, with DPR values exceeding 100%. This trend suggests that, as the nozzle temperature increases, the printed dimensions tend to become larger than the designed dimensions. The upward trend implies that higher temperatures are likely causing over-extrusion, where the material’s increased fluidity leads to excess deposition and consequently larger-than-intended printed parts [27].

In Figure 8B, there is a clear downward linear trend in DPR as the printing speed increases. As the printing speed increases, the DPR decreases, approaching or dropping below 100%. This trend suggests that higher printing speeds may lead to under-extrusion, where the material deposition is insufficient to match the designed dimensions, resulting in parts that are smaller than intended. The decrease in DPR with increasing speed indicates that careful control of printing speed is necessary to prevent dimensional inaccuracies due to inadequate material deposition [26].

In Figure 8C, variations in material feed rate have the most significant impact on DPR, with a strong upward linear trend indicating that increases in material feed rate lead to substantial over-extrusion, causing printed dimensions to exceed the designed specifications. This suggests that precise control of material feed rate is critical for maintaining dimensional accuracy, especially when working with higher material feed rates where the risk of over-extrusion becomes pronounced.

In summary, while nozzle temperature and printing speed contribute to dimensional accuracy, the material feed rate is the primary factor that must be carefully managed to achieve and maintain DPR values close to the ideal 100%. Effective control of material feed rate is essential for ensuring that printed parts adhere to the designed dimensions, minimizing deviations and enhancing overall print quality.

### 3.4. Young’s Modulus

This mechanical parameter measures the stiffness of the scaffold, which should ideally match that of the native bone to support physiological loads without causing stress shielding effects. An appropriate Young’s Modulus is crucial for the scaffold’s structural integrity and its ability to promote load-driven bone regeneration [31].

#### 3.4.1. Influence of Nozzle Temperature on Young’s Modulus

Figure 9A shows a generally decreasing trend in Young’s modulus as the nozzle temperature increases from 180 °C to 250 °C. The modulus begins at approximately 0.392 GPa at 180 °C and decreases to 0.336 GPa at 250 °C, with minor fluctuations. This trend suggests that higher temperatures may reduce the mechanical strength of PLA, potentially due to changes in its microstructure during printing. Thermal degradation or relaxation of internal stresses at elevated temperatures could contribute to this decrease in modulus [32].

The reduction in modulus with increasing temperature can be attributed to the thermomechanical behavior of PLA during the printing process. At higher nozzle temperatures, the viscosity of PLA decreases, improving its flowability. While this enhanced flow allows for better material deposition, it can also weaken interlayer bonding due to excessive spreading before solidification, leading to a lower overall stiffness in the printed scaffold [27].

Cooling dynamics also play a critical role in defining the crystalline structure of printed PLA. Slower cooling, typical at higher nozzle temperatures, results in a more amorphous structure with lower crystallinity. Since higher crystallinity in PLA correlates with greater stiffness due to the crystalline regions’ resistance to deformation, the observed reduction in Young’s modulus at elevated temperatures can be linked to decreased crystallinity from slower cooling rates [33].

This finding is significant for applications where flexibility is more desirable than rigidity, such as in soft tissue engineering. Lower stiffness scaffolds can better mimic the mechanical properties of soft tissues, supporting cell proliferation and integration. In bone healing, selecting a scaffold with an appropriate Young’s modulus can also help mitigate stress shielding, promoting natural bone regeneration [32].

Moreover, nozzle temperature significantly impacts the scaffold’s fatigue performance. Lower temperatures around 190 °C exhibit the highest resistance to fatigue, maintaining structural integrity better under cyclic loading conditions. This is crucial for applications where the scaffold must endure repeated stresses without failing [34,35].

Additionally, higher extrusion temperatures, approaching the upper limit of PLA around 230 °C, enhance the material’s ductility under compression, reduce brittleness, and increase the scaffold’s capacity to bear higher loads without fracturing. These effects highlight the importance of optimizing printing parameters to meet the specific mechanical demands of the intended application [36,37,38].

#### 3.4.2. Impact of Printing Speed on Young’s Modulus

Figure 9B shows that Young’s modulus initially increases with printing speed, peaking at around 2100 mm/min with a modulus of 0.436 GPa, before gradually decreasing as speed continues to rise. This trend suggests that, at moderate printing speeds, there is an optimal balance between material deposition and cooling rates, which enhances mechanical strength. At these speeds, the polymer has sufficient time to cool and partially crystallize before the next layer is deposited, leading to stronger interlayer bonding and higher overall stiffness [39].

However, as printing speed exceeds this optimal point, the layers do not have adequate time to cool and crystallize properly before the next layer is added. This inadequate cooling results in weaker interlayer bonding and reduced crystallinity, both of which contribute to a lower Young’s modulus. Additionally, higher speeds may induce internal stresses within the material due to rapid solidification, further compromising the mechanical properties.

The impact of printing speed on compressive strength is also significant, particularly at lower speeds. Slower speeds improve mechanical interlocking between layers, enhancing the scaffold’s compressive strength. Findings in other studies have indicated that reducing the printing speed from 60 mm/s to 40 mm/s can increase compressive strength by up to 20% [36]. However, compared to other parameters, the influence of printing speed on compressive strength is relatively minor, suggesting that, while important, it should be optimized in conjunction with other parameters for the best overall mechanical performance [40].

#### 3.4.3. Effect of Material Feed Rate on Young’s Modulus

Figure 9C shows a clear positive correlation between material feed rate and Young’s modulus. This relationship suggests that higher material feed rates result in more material being deposited per unit time, potentially leading to denser and mechanically stronger structures, provided that adequate bonding between the layers is maintained.

The importance of material density is further underscored by studies indicating a direct linear relationship between the compressive strength of PLA samples and their density. Higher densities are associated with increased material strength, highlighting the potential to significantly enhance mechanical strength by optimizing printing parameters to improve material density. This observation emphasizes the need to carefully adjust material feed rates to maximize the structural integrity and functional performance of 3D-printed scaffolds [41,42].

In summary, changes in nozzle temperature, printing speed, and layer thickness significantly alter the material’s behavior under compression, which is crucial for designing scaffolds that mimic the mechanical properties of bone [36]. This understanding allows for tailored scaffold designs that can provide the necessary support and stimulation for effective bone healing and integration [43].

Figure 10 compares the influence of various 3D printing parameters—nozzle temperature, printing speed, and material feed rate—on the Young’s modulus of PLA scaffolds by combining and analyzing two parameters simultaneously. The results clearly show that material feed rate plays the dominant role in determining the Young’s modulus. Specifically, as material feed rate increases, the Young’s modulus also increases significantly, suggesting that higher material deposition rates lead to denser scaffolds with improved mechanical properties.

By considering the interplay between these parameters, the figure highlights the potential to optimize scaffold properties by selecting the right combination of printing conditions. For instance, when maintaining a constant material feed rate of 30%, the highest Young’s modulus is achieved within a nozzle temperature range of 200–220 °C and a printing speed range of 1500–2500 mm/min.

These findings provide guidelines for fine-tuning printing parameters to achieve the desired mechanical properties in PLA scaffolds, making it possible to tailor scaffolds for specific clinical or structural requirements. For example, if the goal is to maximize the Young’s modulus while maintaining a specific scaffold porosity, adjusting the nozzle temperature and printing speed within these identified optimal ranges can help achieve that balance.

#### 3.4.4. Relationship Between Geometrical and Mechanical Properties of the Scaffolds

Figure 11 provides a detailed examination of how printing parameters—namely nozzle temperature, printing speed, and material feed rate—affect the dimensional characteristics (height and width) and the mechanical property (Young’s modulus) of PLA scaffolds. This figure is designed to illustrate the relationship between these parameters and the resulting structural and mechanical outcomes of the printed scaffolds.

The lack of a consistent trend between Figure 11A,B compared to Figure 11C shows that, while porosity decreases, the Young’s modulus increases. Figure 11C clearly shows that, as the material density increases (implied by higher material feed rates), the Young’s modulus also increases, highlighting the direct impact of material density on mechanical properties. However, this straightforward relationship is not observed with changes in nozzle temperature or printing speed.

This suggests that the effect of porosity on Young’s modulus is overshadowed by more complex thermal and cooling interactions within the printed material. These factors, which are directly influenced by nozzle temperature and indirectly by printing speed, likely include differential cooling rates, changes in material viscosity, and thermal degradation. The findings indicate that optimizing nozzle temperature and printing speed requires a more nuanced understanding and control, rather than simply attributing changes in Young’s modulus to variations in porosity [35].

In conclusion, while increasing the material feed rate directly enhances the Young’s modulus by reducing scaffold porosity, controlling nozzle temperature and printing speed offers an alternative approach. By carefully adjusting these parameters, it is possible to achieve a slight increase in Young’s modulus while simultaneously allowing for greater scaffold porosity. This balance may be particularly beneficial in applications where both mechanical strength and scaffold porosity are important for supporting tissue regeneration.

#### 3.4.5. Machine Learning Prediction

Figure 12 illustrates the ANN model outputs in comparison to input parameters, showcasing the predicted relationships between nozzle temperature, printing speed, and material feed rate with the corresponding scaffold properties.

The comparative analysis in Table 3 reveals distinct performance characteristics among the modeling approaches. Quadratic polynomial regression demonstrated superior predictive accuracy, achieving the lowest percentage errors (Height: 2.87%, Width: 2.73%, Young’s Modulus: 5.85%), likely due to its explicit incorporation of second-order nonlinear interactions between printing parameters, which effectively captured dominant thermal–mechanical relationships in PLA extrusion. While the ANN exhibited higher errors (Height: 5.13%, Width: 5.52%, Young’s Modulus: 8.79%), its inherent capacity to model high-dimensional and hierarchical nonlinear interactions positions it as a promising candidate for predicting more complex relationships in additive manufacturing, such as multi-physics coupling effects or time-dependent material behaviors, particularly when scaled to larger datasets [44,45]. Future work should explore hybrid approaches combining explicit feature engineering with ANN flexibility, particularly when expanding to multi-scale characterization datasets.

The use of machine learning in this study highlights its potential for accurately predicting the effects of 3D printing parameters, facilitating scaffold customization to meet specific patient needs. The ANN model employed in this research not only optimizes the prediction of maximum Young’s modulus but also ensures that scaffold height and width remain within the critical range of 0.35–0.45 mm, which has porosity of around 42.1% to 59.1%, maintaining appropriate porosity for effective tissue integration [46]. This capability emphasizes the advantage of machine learning in personalized medicine, where customized scaffold designs can significantly enhance patient outcomes by aligning the mechanical properties with individual anatomical and physiological requirements.

The optimal printing parameters identified by the ANN model for maximizing Young’s modulus were a printing temperature of 180 °C, a printing speed of 500 mm/min, and a material feed rate of 29%. Under these conditions, the model predicted a height of 0.371 mm, a width of 0.445 mm (with a porosity of approximately 49.9%), and a Young’s modulus of 0.381 GPa for the scaffold, which falls within the typical range of cancellous bone, usually between 0.1 and 4 GPa [47]. While this value is lower than that of natural healthy cortical bone, it aligns well with the mechanical properties of cancellous bone. This indicates that the scaffold could provide sufficient mechanical support, especially for applications like pelvic cancellous bone defects, where mechanical demands are lower [47,48]. The success of this scaffold demonstrates the effectiveness of the ANN model in optimizing 3D printing parameters, achieving a balance between mechanical strength and geometric requirements. With further refinement in design and material selection, the scaffold could potentially bridge the gap in strength for various bone regeneration applications, highlighting its functionality and structural integrity.

## 4. Conclusions

This study underscores the significant impact of 3D printing parameters on the mechanical properties of PLA scaffolds, particularly Young’s modulus, which is crucial for bone tissue engineering applications. By systematically investigating the effects of nozzle temperature, printing speed, and material feed rate, this research offers valuable insights into optimizing these parameters to improve scaffold performance.

**Nozzle Temperature:** Higher nozzle temperatures reduce Young’s modulus due to thermal degradation and decreased crystallinity, which weaken interlayer bonding.**Printing Speed:** There is an optimal range of printing speeds where Young’s modulus peaks, balancing deposition and cooling rates to enhance crystallinity and interlayer bonding.**Material Feed Rate:** Material feed rate shows a positive correlation with Young’s modulus, as increased material deposition leads to higher material density and improved mechanical strength.**Geometric and Microstructural Factors:** The mechanical performance of PLA scaffolds is influenced by both geometric structure and microstructural characteristics, such as interlayer bonding and crystallization, which are shaped by the printing parameters.**ANN:** The ANN model optimized printing parameters, accurately predicting the maximum Young’s modulus while adhering to geometric constraints. The model demonstrated the ability to balance mechanical strength and scaffold geometry, identifying the optimal printing temperature, speed, and material feed rate. Notably, the Young’s modulus achieved falls within the typical range for cancellous bone, indicating the model’s potential to meet specific clinical requirements, particularly in cases where mechanical demands are lower.

Future work could focus on expanding the dataset to improve the accuracy and robustness of the model. With a larger and more diverse dataset, the predictive capability of the model could be further refined, allowing for better control of printing parameters and the development of patient-specific scaffolds that closely mimic the mechanical properties of natural bone, thereby enhancing the effectiveness of PLA scaffolds in clinical applications.

## Figures and Tables

**Figure 1 bioengineering-12-00315-f001:**
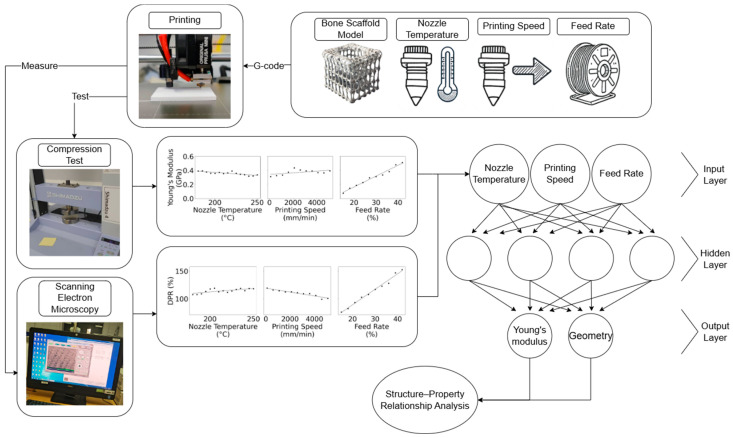
Experimental process for using an ANN to analyze the impact of printing parameters on the geometric and mechanical properties of PLA scaffolds.

**Figure 2 bioengineering-12-00315-f002:**
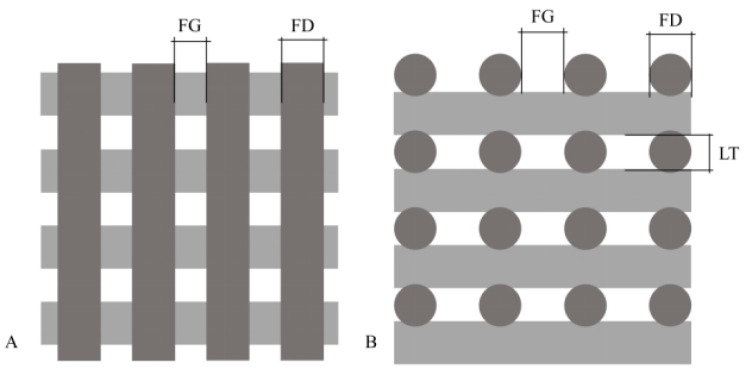
The top (**A**) and side (**B**) views of the scaffold.

**Figure 3 bioengineering-12-00315-f003:**
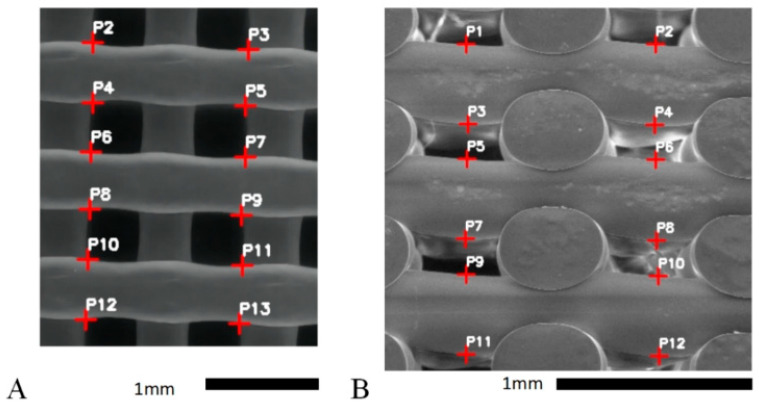
SEM images of bone scaffolds from (**A**) top view and (**B**) side view.

**Figure 4 bioengineering-12-00315-f004:**
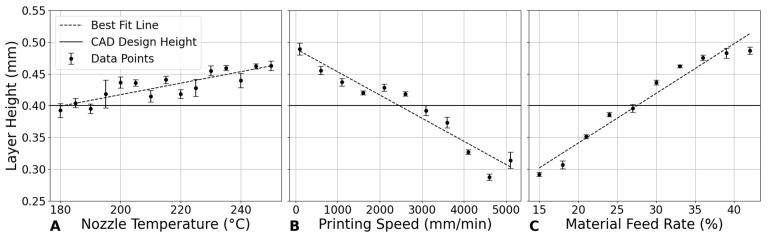
Impact of printing parameters on layer height. (**A**) Nozzle temperature at a constant printing speed of 800 mm/min and material feed rate of 30%. (**B**) Printing speed at a constant nozzle temperature of 210 °C and material feed rate of 30%. (**C**) Material feed rate at a constant nozzle temperature of 210 °C and printing speed of 800 mm/min.

**Figure 5 bioengineering-12-00315-f005:**
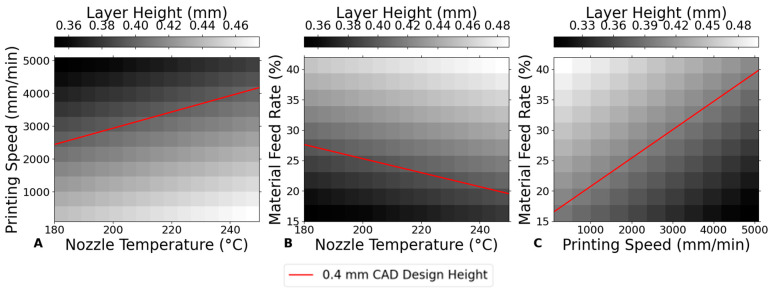
Impact of printing parameters on layer height. (**A**) Nozzle temperature and printing speed at constant material feed rate of 30%. (**B**) Nozzle temperature and material feed rate at constant printing speed of 800 mm/min. (**C**) Printing speed and material feed rate at constant nozzle temperature of 210 °C.

**Figure 6 bioengineering-12-00315-f006:**
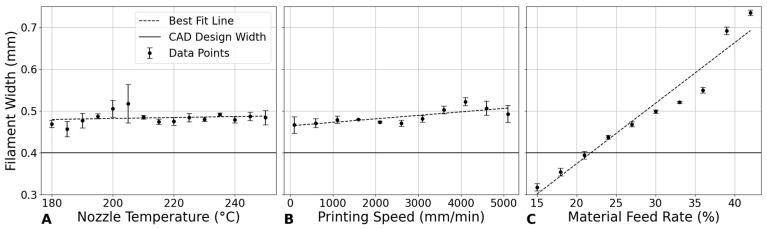
Impact of printing parameters on filament width. (**A**) Nozzle temperature at a constant printing speed of 800 mm/min and material feed rate of 30%. (**B**) Printing speed at a constant nozzle temperature of 210 °C and material feed rate of 30%. (**C**) Material feed rate at a constant nozzle temperature of 210 °C and printing speed of 800 mm/min.

**Figure 7 bioengineering-12-00315-f007:**
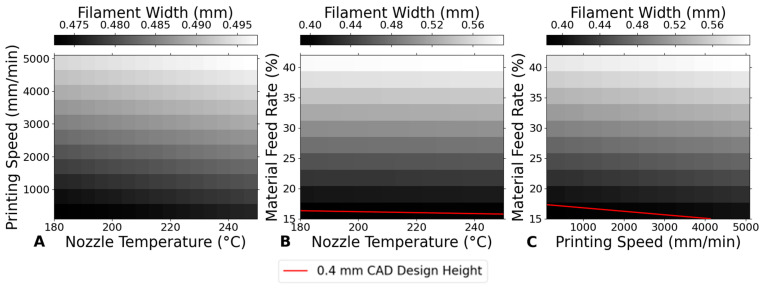
Impact of printing parameters on filament width. (**A**) Nozzle temperature and printing speed at constant material feed rate of 30%. (**B**) Nozzle temperature and material feed rate at constant printing speed of 800 mm/min. (**C**) Printing speed and material feed rate at constant nozzle temperature of 210 °C.

**Figure 8 bioengineering-12-00315-f008:**
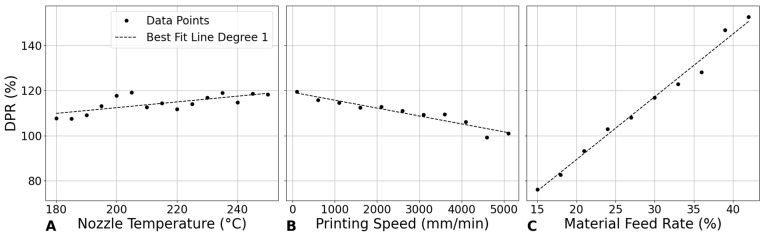
Impact of printing parameters on DPR. (**A**) Nozzle temperature at a constant printing speed of 800 mm/min and material feed rate of 30%. (**B**) Printing speed at a constant nozzle temperature of 210 °C and material feed rate of 30%. (**C**) Material feed rate at a constant nozzle temperature of 210 °C and printing speed of 800 mm/min.

**Figure 9 bioengineering-12-00315-f009:**
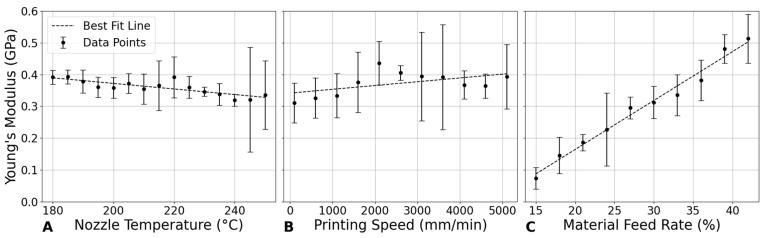
Impact of printing parameters on Young’s modulus. (**A**) Nozzle temperature at a constant printing speed of 800 mm/min and material feed rate of 30%. (**B**) Printing speed at a constant nozzle temperature of 210 °C and material feed rate of 30%. (**C**) Material feed rate at a constant nozzle temperature of 210 °C and printing speed of 800 mm/min.

**Figure 10 bioengineering-12-00315-f010:**
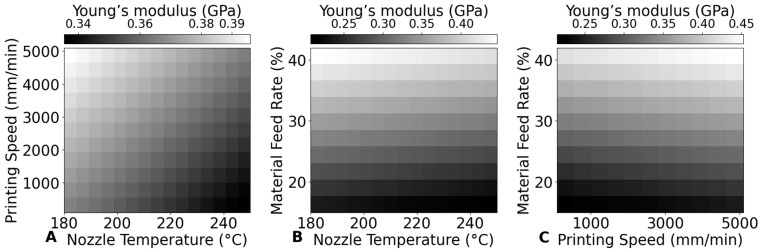
Impact of printing parameters on the Young’s modulus. (**A**) Nozzle temperature and printing speed at constant material feed rate of 30%. (**B**) Nozzle temperature and material feed rate at constant printing speed of 800 mm/min. (**C**) Printing speed and material feed rate at constant nozzle temperature of 210 °C.

**Figure 11 bioengineering-12-00315-f011:**
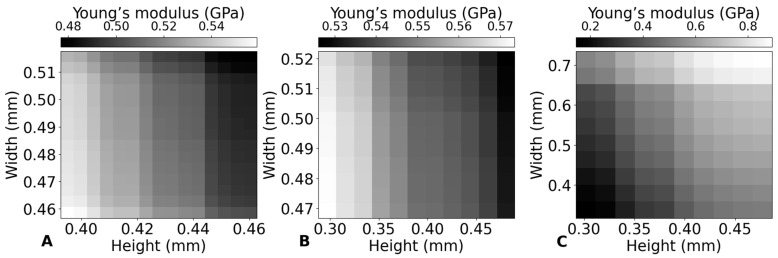
Impact of printing parameters on layer height, filament width and Young’s modulus. (**A**) Nozzle temperature at a constant printing speed of 800 mm/min and material feed rate of 30%. (**B**) Printing speed at a constant nozzle temperature of 210 °C and material feed rate of 30%. (**C**) Material feed rate at a constant nozzle temperature of 210 °C and printing speed of 800 mm/min.

**Figure 12 bioengineering-12-00315-f012:**
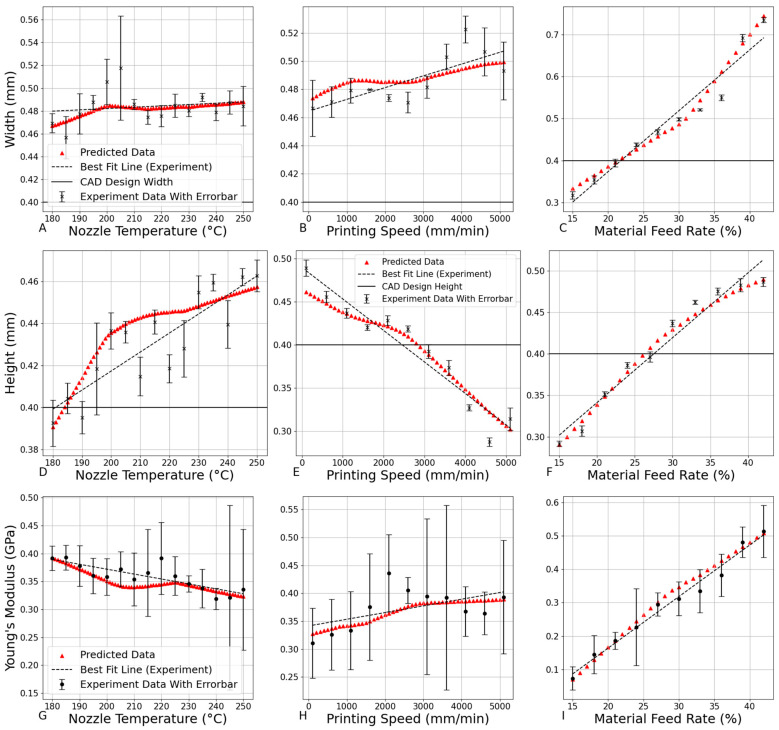
ANN model output vs. input is organized as follows: (**A**–**C**) represent predictions for filament width, (**D**–**F**) represent predictions for layer height, and (**G**–**I**) represent predictions for Young’s modulus. The plots are grouped by 3D printing parameters: (**A**,**D**,**G**) show the effects of nozzle temperature, with a constant printing speed of 800 mm/min and a material feed rate of 30%. (**B**,**E**,**H**) show the effects of printing speed, with a constant nozzle temperature of 210 °C and a material feed rate of 30%. (**C**,**F**,**I**) show the effects of material feed rate, with a constant nozzle temperature of 210 °C and a printing speed of 800 mm/min.

**Table 1 bioengineering-12-00315-t001:** Design of Experiment Parameters.

Parameter	Range	Increment	Number of Groups
Printing Temperature	180 °C to 250 °C	5 °C	15
Printing Speed	100 mm/min to 5100 mm/min	500 mm/min	11
Material feed rate	15% to 42%	3%	10

**Table 2 bioengineering-12-00315-t002:** ANOVA Results.

Parameter	Response Variable	F-Value	*p*-Value
Temperature	Young’s Modulus	3.41	2.36 × 10^−3^
	Height	8.98	3.09 × 10^−7^
	Width	9.06	2.78 × 10^−7^
Printing Speed	Young’s Modulus	3.93	3.56 × 10^−3^
	Height	234.85	5.80 × 10^−20^
	Width	6.69	1.05 × 10^−4^
Material feed rate	Young’s Modulus	15.67	3.15 × 10^−7^
	Height	361.49	3.89 × 10^−20^
	Width	781.15	1.83 × 10^−23^

**Table 3 bioengineering-12-00315-t003:** Comparison of Prediction Errors.

Model	Height Predictions Errors (%)	Width Predictions Errors (%)	Young’s Modulus Predictions Errors (%)
Linear Regression	3.65	3.89	8.15
Quadratic Polynomial Regression	2.87	2.73	5.85
ANN	5.13	5.52	8.79

## Data Availability

The data supporting the findings of this study are available from the corresponding author upon reasonable request.

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
