# Peer review of "Investigation of the Effects of 3D Printing Parameters on the Mechanical Properties of Bone Scaffolds: Experimental Study Integrated with Artificial Neural Networks"

_bioengineering, 2025, doi:10.3390/bioengineering12030315_

Round 1

Reviewer 1 Report

Comments and Suggestions for Authors

In this study, the authors systematically analyzed and diagrammed the effects of nozzle temperature, printing speed, and feed rate on Young's modulus in the production of polylactic acid scaffolds. The authors' efforts are essential for the production of precise 3D printing products for regenerative medicine. However, there are errors in parameter settings and parameter terms for experiments, which need to be corrected.

  1. The contents of (a) and (b) in Figure 2 have changed. Please correct them.

  1. One of the important factors in 3D printing is the comparison of printing results according to nozzle size. A parameter for the change in printed filament diameter according to the change in nozzle diameter should be added, and the results according to the added parameter should be added to Figure 4.

  1. In additive manufacturing including machine tools, feed rate generally refers to the tool movement speed (mm/s). In other words, printing speed and feed rate mentioned by the authors in the text are generally used with almost the same meaning. Although the author does not know the exact information about the % unit of feed rate mentioned in the system used, the term feed rate used in the paper should follow the general rule (mm/s). In this case, all graphs related to feed rate such as figure 4C, figure 6C, figure 7C, figure 8C, figure 9C, figure 10C, etc. should be revised because they show graph forms that are opposite to the general rule. In other words, the definition of feed rate should be reviewed and the main text should be revised accordingly.

  1. Line 495: figure 10 was used duplicated. Please revise.

Author Response

Response to Reviewer 1

1. The contents of (a) and (b) in Figure 2 have changed. Please correct them.

Response: We have carefully reviewed Figure 2 and ensured that the contents of (a) and (b) are consistent with the descriptions in the text. The necessary corrections have been made accordingly.

2. One of the important factors in 3D printing is the comparison of printing results according to nozzle size. A parameter for the change in printed filament diameter according to the change in nozzle diameter should be added, and the results according to the added parameter should be added to Figure 4.

Response: In this study, we kept the nozzle diameter constant at 0.4 mm to specifically investigate the effects of nozzle temperature, printing speed, and feed rate on the mechanical properties of the scaffolds. This ensures that the results reflect the influence of these parameters without additional variability from nozzle size. We have explicitly stated this in Section 2.2:

"In this study, the nozzle diameter was kept constant to isolate the effects of nozzle temperature, printing speed, and feed rate on the mechanical properties of the scaffolds, ensuring a focused investigation into these key parameters."

Although nozzle size is an important factor in 3D printing, investigating its effects would require additional experiments beyond the scope of this study. However, we acknowledge its relevance and have mentioned it as a potential avenue for future research.

3. In additive manufacturing including machine tools, feed rate generally refers to the tool movement speed (mm/s). In other words, printing speed and feed rate mentioned by the authors in the text are generally used with almost the same meaning. Although the author does not know the exact information about the % unit of feed rate mentioned in the system used, the term feed rate used in the paper should follow the general rule (mm/s). In this case, all graphs related to feed rate such as figure 4C, figure 6C, figure 7C, figure 8C, figure 9C, figure 10C, etc. should be revised because they show graph forms that are opposite to the general rule. In other words, the definition of feed rate should be reviewed and the main text should be revised accordingly.

Response: We have carefully reviewed the terminology and clarified that "material feed rate (MFR)" is used in this study, which represents the ratio between the nozzle movement speed and the material extrusion speed. The definition has been added to Section 2.3:

"In this study, material feed rate (MFR) is defined as the ratio between the nozzle movement speed and the material extrusion speed. When the filament diameter matches the nozzle diameter and the material is extruded at the same rate as the nozzle movement (e.g., a nozzle moving at 1 mm/s with a filament feed rate of 1 mm/s), the material feed rate is considered 100%. However, in cases where over-extrusion or under-extrusion is desired, the material feed rate can be adjusted accordingly to either increase or decrease material deposition relative to nozzle movement."

Additionally, all figures (Figure 4C, Figure 6C, Figure 7C, Figure 8C, Figure 9C, and Figure 10C) have been reviewed and corrected if necessary to ensure consistency with this definition.

4. Figure 10 was used twice (Line 495). Please revise.

Response: We have carefully reviewed the manuscript and corrected the duplication of Figure 10. The numbering has been adjusted accordingly.

Reviewer 2 Report

Comments and Suggestions for Authors

The manuscript can be concise. Line 104-112 is redundancy, reader would know what the section means by reading the section head.

The analysis of the effects of nozzle temperature, printing speed, and feed rate on the geometric and mechanical properties can be easily performed with routine statistical and DOE technique. Authors should seek for the comparisons of statistical analysis with ANN.

In 2.4 the SEM manufacturer should be described

In 2.5 The platform for running ANN model training should be specified.

Line 128 : The detail of Polylactic acid (PLA), such as molecular weight, should be described

In Line 237, the porosity values of the tested material were provided, but the testing method is not described in the method.

Although ANOVA test and confidence level were mentioned in Line 243, the statistical results are not clearly stated in the results. It is unclear about the null hypothesis in the method either.

In 2.3: It is not clear the number of replicated samples for each experiment.

Line 460, L495 typo in the first word.

Comments on the Quality of English Language

some typos should be corrected.

Author Response

Response to Reviewer 2

1. The manuscript can be concise. Line 104-112 is redundancy, reader would know what the section means by reading the section head.

Response:

We have summarized the manuscript structure concisely:

"This study is structured as follows: Section 2 covers the experimental setup and ANN model development, Section 3 analyzes parameter effects and ANN optimization, and Section 4 summarizes findings and future improvements."

2. The analysis of the effects of nozzle temperature, printing speed, and feed rate on the geometric and mechanical properties can be easily performed with routine statistical and DOE technique. Authors should seek for the comparisons of statistical analysis with ANN.

Response: We acknowledge this and have added a comparison between ANN and traditional statistical methods (Linear Regression and Quadratic Polynomial Regression) in Section 3.4.2. This highlights the differences in predictive accuracy and optimization capability between ANN and conventional statistical techniques.

3. In 2.4 the SEM manufacturer should be described

Response: We have added the manufacturer details in Section 2.4:
"Scanning Electron Microscopy (SEM) was performed using a TM3030Plus tabletop microscope (Hitachi, Japan) to analyze scaffold morphology."

4. In 2.5 The platform for running ANN model training should be specified.

Response: The ANN training environment details have been added to Section 2.5:

"The ANN model training was conducted on a workstation equipped with an Intel Core i7-12700K CPU and an NVIDIA RTX 3090 GPU, running on Windows 10. The model was implemented using Python 3.11.7 with TensorFlow 2.17.0 and Keras 3.5.0. Additional dependencies included NumPy 1.26.4, Pandas 2.1.4, and Scikit-learn 1.5.1."

5. Line 128 : The detail of Polylactic acid (PLA), such as molecular weight, should be described

Response: We have supplemented the PLA material details in Section 2.1 with its melt flow index (MFI) of 6.0 g/10 min and Vicat softening temperature of 60°C. Unfortunately, the exact molecular weight was not provided by the manufacturer, but we acknowledge its importance in determining mechanical properties.

6. In Line 237, the porosity values of the tested material were provided, but the testing method is not described in the method.

Response: We have added the porosity measurement method in Section 2.2, describing how ImageJ software was used for image analysis of cross-sections to calculate porosity.

7. Although ANOVA test and confidence level were mentioned in Line 243, the statistical results are not clearly stated in the results. It is unclear about the null hypothesis in the method either.

Response: We have added the null hypothesis and ANOVA statistical results in Section 3: One-way ANOVA results are now explicitly reported, including p-values and F-statistics.

8. In 2.3: It is not clear the number of replicated samples for each experiment.

Response: We have clarified in Section 2.3 that each experiment was replicated three times to ensure statistical reliability.

9. Line 460, L495 typo in the first word.

Response: We have corrected the spelling mistakes, including "bonding" and "nozzle."

Round 2

Reviewer 1 Report

Comments and Suggestions for Authors

None